# How can we assess human-agent interactions?
# Case studies in software agent design

**Valerie Chen** [1]  **Rohit Malhotra** [2]  **Xingyao Wang** [2]  **Juan Michelini** [2]  **Xuhui Zhou** [1]  **Aditya Bharat Soni** [2]
**Hoang H. Tran** [2]  **Calvin Smith** [2]  **Ameet Talwalkar** [1]  **Graham Neubig** [1,2]

## Abstract

While benchmarks measure the accuracy of LLM-powered agents, they mostly assume full automation, failing to represent the collaborative nature of real-world use cases. In this paper, we make two major steps towards the rigorous assessment of human-agent interactions. First, we propose `PULSE`, a framework for more efficient human-centric evaluation of agent designs, which comprises collecting user feedback, training an ML model to predict user satisfaction, and computing results by combining human satisfaction ratings with model-generated pseudo-labels. Second, we deploy `PULSE` in software engineering—one of the highest-impact, real-world domains for LLM agents—via a large-scale web platform built around the open-source agent OpenHands. Across 15k users, we evaluate how three agent design decisions impact developer satisfaction rates. We also show how `PULSE` can lead to more robust conclusions about agent design, reducing confidence intervals by 40% compared to a standard A/B test. Finally, we find substantial discrepancies between in-the-wild results with benchmark performance (e.g., the anti-correlation between `claude-sonnet-4` and `gpt-5`), underscoring the limitations of benchmark-driven evaluation. Our framework `PULSE` provides guidance for future evaluations, and our findings identify opportunities for better software agent designs.

## 1. Introduction

Agents are simultaneously one of the most promising emerging technologies empowered by LLMs ([QuantumBlack & Technology, 2025](#)), and a perfect storm of complexity and unpredictability for the AI researchers and engineers who are tasked with creating them. There are a plethora of design decisions that any agent developer must face, such as which underlying language model to use ([Yue et al., 2025](#)), what tools to provide to the agent ([Jin et al., 2025](#); [Soni et al., 2025](#)), how to prompt the agent to use its capabilities effectively ([Khattab et al., 2023](#); [Spiess et al., 2025](#)), and how to plan and coordinate across tasks or sub-workflows ([Fourney et al., 2024](#)). Errors in any of these areas can reduce the agent's effectiveness or lead to performance regressions in deployed systems ([Cemri et al., 2025](#)).

Our current best tool for diagnosing and improving agent performance is a rigorous measure of accuracy on agent benchmarks. Fortunately, given the importance and interest in agents, there is now a variety of benchmarks that can be used across areas, such as software engineering ([Jimenez et al., 2023a](#); [Yang et al., 2024](#); [Zan et al., 2025](#)), web browsing ([Zhou et al., 2023](#); [Koh et al., 2024](#)), and scientific discovery ([Chen et al., 2024](#)). On the other hand, these benchmarks are largely based on the premise of *full task automation*, where the agent finishes a well-specified task with no user feedback. Though some have claimed that agents will be eventual replacements for large swaths of human work ([Shibu, 2025](#)), in reality, current agentic systems work closely together with human supervisors to *complete tasks collaboratively*. While there has been some attempts to simulate human interaction ([Vijayvargiya et al., 2026](#); [Pan et al., 2025](#)), to our knowledge, there has not, to date, been a rigorous evaluation protocol proposed or empirical results presented in this setting.

In this paper, we make two major steps towards *rigorous assessment of human-agent interactions*. **First, we propose `PULSE`—Prediction-powered User Label Synthesis and Evaluation—a three-step framework for measuring the effect of a proposed agent chang** (Figure 1, top). First, we set up the interface and data collection mechanism for user feedback from human-agent interactions. We then train an ML model to predict user satisfaction by extracting important features about the user, agent, and task completion status. Finally, we extend prediction-powered inference ([Angelopoulos et al., 2023b](#)) to provide valid confidence inter-

[1]Carnegie Mellon University [2]OpenHands. Correspondence to: Valerie Chen <valeriechen@cmu.edu>.

*Proceedings of the $43^{rd}$ International Conference on Machine Learning*, Seoul, South Korea. PMLR 306, 2026. Copyright 2026 by the author(s).

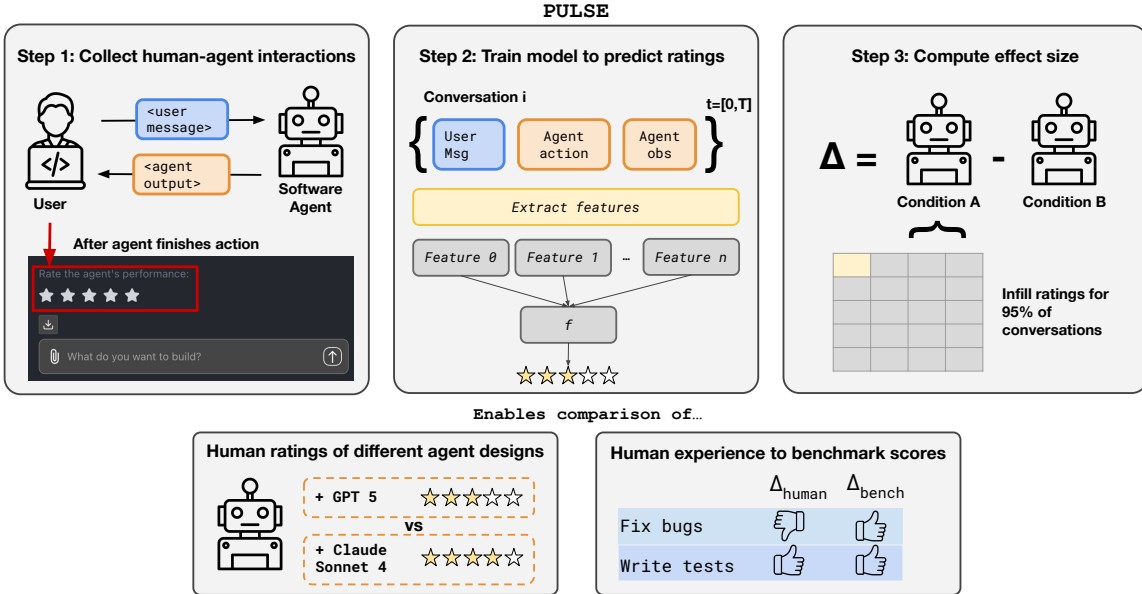

*Figure 1.* **Our framework `PULSE` for efficient human-centric evaluation of agent designs enables insights into how user experience varies with agent design and comparison to benchmark performance.** We instantiate `PULSE` in a software engineering agent use case and conduct a set of case studies to identify novel insights into designing useful, collaborative agents.

vals about the effect size of a proposed agent change. In our case studies, we find that `PULSE` can reduce confidence interval widths by an average of nearly 40% compared to a standard A/B test.

**Second, we deployed `PULSE` in the wild to conduct the first large-scale evaluation of agent designs, in software engineering application** (Figure 1, bottom). We focus on software agents as they are arguably the most commercially impactful use case of current agents (Github, 2022; Cursor, 2023; Cognition, 2024; Wang et al., 2024; Anthropic, 2023). Further, software engineering is also one of the most developed domains for agent benchmarking, making it an ideal testbed for comparing benchmark performance to in-the-wild human feedback. We use a web-based platform where users perform day-to-day coding tasks with Open-Hands (Wang et al., 2024), an open-source, state-of-the-art software engineering agent. Across over 36k sessions and 15k users, we conducted three case studies that varied the choice of LLM backbone and scaffolding changes, like the planning strategy and memory mechanism.

**Finally, we show `PULSE` identifies impactful agent design choices and captures where benchmarks fall short.** Our results show that investing in stronger base models yields large, statistically significant changes in user satisfaction (i.e., $\Delta = 6\text{-}8\%$). While scaffolding changes had relatively less impact (i.e., $\Delta < 3\%$), we still observe benefits of showing users the agent's plan and find that changing memory parameters can also lead to cost savings without degradation to user experience. We also compare our re-

sults to 7 different code-related benchmarks and find that differences in models across benchmarks do not necessarily translate into human ratings. In particular, while `gpt-5` outperforms `claude-sonnet-4` on 6 out of 7 benchmarks, humans prefer `claude-sonnet-4` over `gpt-5` on 4 out of the 7 task subsets. In summary, our findings highlight the importance of human-in-the-loop evaluation.

Software engineering agents serve as a high-impact testbed in this work, but we believe `PULSE` can be applied as a general framework for evaluating human–agent collaboration. Although raw code contexts cannot be shared for privacy reasons, we will release code for the full `PULSE` framework and anonymized feature-level datasets, enabling researchers to reproduce the modeling and statistical components of our work and apply them to their own deployments. In Appendix C, we also discuss how our findings motivate better agent designs and how `PULSE` can be readily adapted to other domains.

## 2. Related Work

**Coding agent evaluation.** Evaluation of coding agents has become a major focus in both ML and software engineering, yet existing approaches remain dominated by static, task-level benchmarks (Jimenez et al., 2023a; Yang et al., 2024; Zan et al., 2025). Recent benchmarks broaden coverage to diverse software engineering tasks, including test generation, repository-level reasoning, and fixing continuous integration failures (Mündler et al., 2024; Bogomolov et al., 2024), and a smaller line of work introduces interactive or

| | Real Users | Vary Agent Design | In-the-Wild Tasks | Links to Benchmark |
|---|---|---|---|---|
| Static benchmarks | ✗ | ✓ | ✗ | ✓ |
| Interactive benchmarks | ✗ | ✓ | ✗ | ✓ |
| Prior human studies | ✓ | ✗ | ✗ | ✗ |
| **Our Work** | ✓ | ✓ | ✓ | ✓ |

*Table 1.* **Comparison of coding agent evaluation paradigms.** Prior work studies either controlled benchmarks or real users, but not both while varying agent design and connecting outcomes back to benchmark performance.

multi-step environments with simulated users (Vijayvargiya et al., 2026; Pan et al., 2025), but these settings still operate under controlled, benchmark-driven conditions. Human-in-the-loop studies of coding systems primarily compare traditional copilots with newer agentic workflows and focus on productivity or usage outcomes (Anthropic, 2025c; Chen et al., 2025); however, they typically evaluate a single system and do not examine how different agent design choices affect performance. We address this gap by studying coding agents deployed in the wild and developing methods to run efficient, large-scale human-in-the-loop experiments in these settings, linking in-the-wild outcomes to benchmark performance. To our knowledge, this is the first large-scale study of deployed coding agents that jointly vary agent design, measure real user impact, and connect these findings back to benchmark evaluation (Table 1).

**User satisfaction estimation.** Prior works in the speech and dialogue communities have explored user satisfaction in multi-turn chat interactions using signals which include thumbs up/down ratings and a 5-point satisfaction scale (Sun et al., 2021a). Methodologies for predicting user satisfaction range from using text embeddings (Liang et al., 2021) to more recent LLM-powered approaches (Hu et al., 2023; Lin et al., 2024). Unlike dialogue settings where turns consist purely of text exchanges, our work studies predicting human satisfaction in *agent* settings, where agent trajectories couple language with state-changing actions, tool invocations, and environment observations. We find that standard LLM-based approaches that excel in dialogue settings struggle in our setting and that our predictive methods outperform these baselines.

**Efficient estimation of effect size with noisy samples.** From clinical trials to public health interventions, there is growing interest in running controlled trials of new treatments more quickly (De Bartolomeis et al., 2025; Poulet et al., 2025; Demirel et al., 2024). One statistical machinery that makes this possible is prediction-powered inference (PPI), which is an approach that reduces the variance of estimators by leveraging predictions from a prediction model $f$ on unlabeled data by constructing a correction term using the labeled data (Angelopoulos et al., 2023a;b). Prior work has extended PPI to boost the efficiency of experiments by generating a "digital twin" (i.e., the counterfactual) in randomized control trials (Poulet et al., 2025) or creating synthetic examples that are then labeled using an LLM (Shankar & Fiterau, 2024). Our setting differs from prior work, where there is no obvious choice for $f$ for evaluating human-agent interactions. We discuss how to train such a model for handling human-agent trajectories and use PULSE in a series of real-world evaluations.

## 3. Methods

We overview PULSE, our evaluation framework to efficiently compare agent designs overviewed in Figure 1: (1) feedback data collection, (2) training an ML model to predict user satisfaction, and (3) computing test results and confidence intervals. We use software agent design to instantiate each step, with additional details provided later in Section 4.

### 3.1. Feedback Data Collection

Collecting user reviews and ratings has been a long-standing practice to understand product quality and guide iterative improvement (McAuley et al., 2012; Fabijan et al., 2015); similarly, to evaluate different agent designs, we ask users for feedback on how they perceived the agent to have performed. However, in the context of human-agent interactions, a natural question is when we should ask for feedback. We want feedback collection to be minimally invasive and align with the agent's own sense of task completion.

We propose a design where, in the chat window where users and agents communicate, users are prompted to provide feedback after each *work segment*. This is similar to how you would provide a rating for a completed ride in a ridesharing app or a finished purchase on an online marketplace. A work segment comprises all the events that unfold between when the user sends a command, the agent enters a "running" state, and returns to "stopped". At this point, the chat interface shows the user text that says "rate the agent's performance" and asks the user to provide a rating on 5 star scale. Asking for feedback right after a work segment grounds the user's rating in a specific, just-finished task, which helps avoid noisy or unfocused judgments.[1] We

---

[1]We note that implicit signals (e.g., dwell time, edit behavior) are a potential extension beyond explicit ratings. However, prior studies show such signals may not consistently correlate with satisfaction ratings (Joachims et al., 2007; Jeunen, 2019).

*Figure 2.* **Feedback Collection Interface.** After each work segment—i.e., when a user sends a command and an agent has done work to complete that command, we ask the user to provide a rating of the agent's performance.

provide an example of the interface in Figure 2.

In summary, we define each work segment $i$ as $W_i = \{M_i, T_i, Y_i\}$ where the trajectory $T_i = \{a_{i,1}, o_{i,1}, a_{i,2}, \dots\}$ is a list of actions and observations from the agent and $Y_i$ is the user rating (which may be $\emptyset$ because the user did not choose to give feedback). As such, each session $X_i = \{W_1, \dots, W_j\}$ can consist of one or more work segments. If there are multiple ratings, we take the average across segments $\bar{Y}_i$, which provides us with more granular ratings than the 5 star scale. Across many sessions, we can create a dataset of human-agent interactions and user ratings $\mathcal{D} = \{(X_i, \bar{Y}_i)\} \cup \{(\tilde{X}_i, \emptyset)\}$ where $X_i$ are the sessions where there is at least one rating and $\tilde{X}_i$ are the ones without. We next describe how we train a model to infill missing ratings.

### 3.2. Predicting Human Satisfaction

Given a dataset of labeled trajectories $\mathcal{D} = \{(X_i, \bar{Y}_i)\}$, we train a ML model $f$ to infill user satisfaction $\bar{Y}_i$, when it is $\emptyset$, given human-agent interaction in the session $X_i$. Unlike prior work on predicting satisfaction in multi-turn dialogue (Hu et al., 2023; Lin et al., 2024), a unique challenge with handling agent trajectories is balancing the number of labeled samples with the dimensionality of agentic trajectories (i.e., each $T_i$ is a complex object that may easily comprise tens thousand tokens).

**Identifying feature set.** We developed the feature set through an iterative human-in-the-loop process. We randomly sampled collected trajectories, then prompted frontier LLMs (e.g., o3, claude-4-opus) to identify behavioral patterns distinguishing high versus low-rated interactions. Multiple domain experts reviewed candidate features, merging overlapping categories, splitting overly broad ones, and refining definitions for consistent annotation. We iterated until the set of features stabilized. In Appendix A.2, we explore how to use LLMs (without human-in-the-loop) to

automatically discover the majority of these features.

**Feature categories.** The annotation process yielded 15 features that can be grouped into three categories. We also ground the categories in prior literature, which we reference for each category below.

- *Features based on the user.* These features include user messages $\{M_i\}$, which can convey information about how satisfied a user is with the agent, and user sentiment. Additionally, the number of messages that users choose to send $|\{M_i\}|$ is also included. We note these features have also been considered in information retrieval and dialogue literature (Sun et al., 2021b; Lin et al., 2024).

- *Features based on the agent.* These features indicate what kind of task the user is working on (e.g., implementing new features or fixing bugs), which can be detected through the user messages and agent trajectory, as well as potential failure modes of the agent (e.g., insufficient testing or did not follow instruction). We can consider these features as adaptations from task-oriented dialogue (Higashinaka et al., 2015; Siro et al., 2022).

- *Features that show task progression.* In software engineering, task progression is often signaled through git actions (Saidani et al., 2020). For example, if $T_i$ contains actions where the agent pushes code towards the end of a session may be a sign of user satisfaction with the agent's work.

**Training a prediction model.** Using $\mathcal{D}$, we train various ML models ranging from logistic regressions to random forests. We also compare to a baseline of simply providing the full, raw trajectory $X_i$ to an LLM-as-a-judge (Zheng et al., 2023) without the feature post-processing, using models that can handle particularly long contexts (e.g., o3 and gemini-2.5-pro). In Section 6, we compare various model performances on our software agent use case.

### 3.3. Comparing agent designs

With a feedback collection mechanism and labeling model $f$ in hand, we discuss how to efficiently compare agent designs. To compare different agent designs, we adopt an A/B testing framework, a widely used approach in fields such as human-computer interaction and web experimentation to compare different system variants (Siroker & Koomen, 2015). Each test produces datasets indexed by condition, $\mathcal{D}_c = \{(X_i, Y_i)\}$ where $c$ denotes the different versions of the agent being compared.

**Naive effect size estimation.** We are interested in estimating the true difference in average user satisfaction between

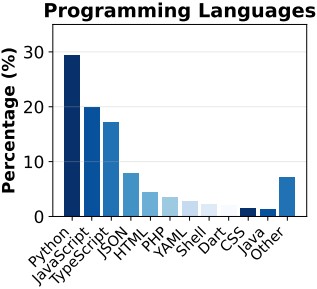
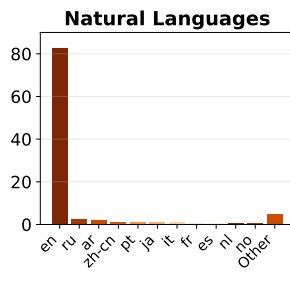
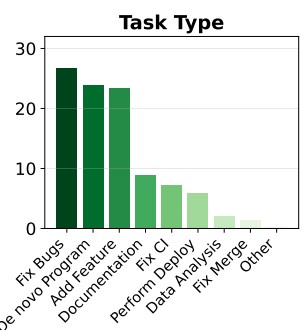
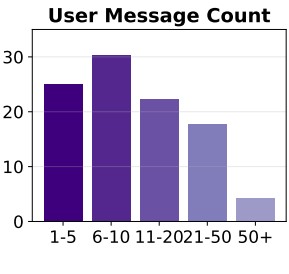

*Figure 3.* **Overview of statistics of in-the-wild deployment of our human-agent evaluation framework.** Our evaluations spanned over 15k users who were working on a diverse set of problems in terms of programming and natural languages, task category, and interaction style with the agent (as seen through user message count).

conditions:

$$\widehat{\Delta}_{\text{naive}} = \frac{1}{n_{c_1}} \sum_{i \in c_1} Y_i \; - \; \frac{1}{n_{c_2}} \sum_{i \in c_2} Y_i$$

Given $\mathcal{D}_{c_1}$ and $\mathcal{D}_{c_2}$, we compute the empirical difference $\widehat{\Delta}_{\text{naive}}$ and test whether it is statistically significant from 0 (i.e., no difference between $c_1$ and $c_2$) using a bootstrap permutation test.

**Augmenting with unlabeled trajectories.** However, since we have $f$, we can directly extend prediction-powered inference (PPI) (Angelopoulos et al., 2023a;b) to construct an effect size estimator. Note that PPI does not require $f$ to be unbiased or perfectly accurate; systematic prediction error is corrected using labeled data, as long as labeled and unlabeled trajectories are drawn from the same underlying distribution. PPI provides the recipe for the mean estimator of each condition $c$ with tuning parameter $\lambda_c \in \mathbb{R}$ given $n_c$ labeled and $N_c$ unlabeled trajectories:

$$\widehat{\mu}_c(\lambda_c) = \underbrace{\frac{1}{n_c} \sum_{i=1}^{n_c} Y_i}_{\text{sample mean of labels}}$$

$$+ \lambda_c \left( \frac{1}{N_c} \sum_{j=1}^{N_c} \underbrace{f(\tilde{X}_j)}_{\text{unlabeled traj.}} - \frac{1}{n_c} \sum_{i=1}^{n_c} \underbrace{f(X_i)}_{\text{labeled traj.}} \right).$$

The optimal $\lambda_c$ can be computed using a sample plug-in comprising $\widehat{\text{Cov}}(Y, f(X)|Z = c_i)$ as well as the variance of $f$. Accordingly, the augmented effect size estimator is the difference

$$\widehat{\Delta}_{\text{augment}} = \widehat{\mu}_{c_1}(\widehat{\lambda}_{c_1}) - \widehat{\mu}_{c_2}(\widehat{\lambda}_{c_2}).$$

Under the regularity conditions in PPI, $\sqrt{n_c}(\widehat{\mu}_b(\lambda_c) - \mu_c) \overset{d}{\to} \mathcal{N}(0, \sigma_c^2(\lambda_c))$. Because trajectores from each condition (and their unlabeled pools) are independent conditional on assignments, each estimator is asymptotically

independent. A Wald confidence interval is therefore

$$\widehat{\Delta}_{\text{augment}} \pm \; z_{1-\alpha/2} \sqrt{\frac{\widehat{\sigma}_{c_1}^2(\widehat{\lambda}_{c_1})}{n_{c_1}} + \frac{\widehat{\sigma}_{c_2}^2(\widehat{\lambda}_{c_2})}{n_{c_2}}},$$

where each $\widehat{\sigma}_b^2(\widehat{\lambda}_b)$ is the plug-in estimate of $\sigma_b^2(\cdot)$.

## 4. Deployment Details

Software agents are the focus of this work, as it is arguably the most commercially impactful use case of current LLM-powered agents. There are many coding agents available, including Devin (Cognition, 2024) and Claude Code (Anthropic, 2025a), but they are largely closed-source. We performed our study using OpenHands, a leading open-source coding agent, as measured by SWE-Bench (Jimenez et al., 2023b) and other benchmarks. All users opted in to the collection and analysis of statistical data, such as the data gathered in this study.

**Summary of user data.** In total, our study comprised over 36k sessions from 15k different users. From these sessions, we obtained $N = 1747$ labeled trajectories (mean rating 4.07) and roughly $20\times$ more unlabeled sessions, as only about 5% of interactions receive ratings. We find that only 12.75% of users contributed ratings in multiple sessions (and of that percentage, the majority contributed only 2 sessions). The vast majority of users only contributed one session. Figure 3 overviews the distribution of agent usage across multiple features: Across these users, they worked on a diverse set of tasks. When we classify a sample of the trajectories, we find that the majority of users were trying to `fix bugs` and `create programs from scratch`. There were multiple programming languages, with Python being the most popular (29.52%) and predominantly English-speaking users (82.61%). Finally, users sent a median of 10 messages to the agent in a session.

**Comparing labeled and unlabeled trajectories.** We empirically compared trajectories across all 15 features used

in our predictive model and quantified the differences using rank-biserial correlation. Overall, we found that only the user message count exhibits a moderate effect, with labeled trajectories containing more user messages (RBC=0.32). All remaining differences between labeled and unlabeled trajectories are small or negligible (RBC around 0.1 or less). The primary difference in user message count suggests that explicit feedback tends to be provided by more verbose or engaged users, but unlabeled trajectories still exhibit similar patterns of agent failures and user sentiment, indicating that silent users likely experience similar issues but may not choose to provide explicit ratings. We note that although only a small fraction of users provide ratings, these responses still come from nearly 1k unique users.

**Are low user ratings meaningful indicators of other issues?** When correlating user satisfaction and more objective interaction metrics (e.g., git actions), we find that satisfaction is positively correlated with git push ($r = 0.117$, $p < 0.001$) and git commit ($r = 0.101$, $p < 0.001$), and near zero for other git actions. While these effects are small, they indicate alignment between subjective ratings and objective outcomes. Additionally, we manually inspected the event stream of a sample of 20 sessions that had low ratings and found that features like user sentiment and volume of user messages are often downstream of concrete interaction failures rather than purely attitudinal signals. For example, in these low-rated sessions, we observed that there are repeated errors like failed tests/CI, missing dependencies, and port/health-check failures. In these sample sessions, we see that the user can get increasingly frustrated (based on the tone and content of their messages) after spending multiple turns trying to recover from these failures. This observation suggests that higher message volume and negative sentiment frequently reflect interaction friction, which then drives satisfaction down. As such, we explore what agent designs can improve low user ratings in Section 6.

## 5. Experimental Design

### 5.1. Overview of case studies

Beyond the LLM model that serves as the agent's backbone, prior work has discussed the importance of other aspects like planning, memory, and tool use (Weng, 2023; Sumers et al., 2023; Durante et al., 2024). We apply PULSE to study three aspects of software agent design.

**Case Study 1: LLM Model.** The LLM model backbone is arguably one of the most important components of an agent and is the primary independent variable reported when reporting results on benchmarks. We compared 3 different state-of-the-art models in terms of agentic coding performance: `claude-3.7-sonnet`, `claude-4-sonnet`, and `gpt-5` (reasoning effort high). We report

results of two separate tests, the first, which compared `claude-3.7-sonnet` and `claude-4-sonnet`, and the second, which compared `claude-4-sonnet` and `gpt-5`. All other aspects of the agent design are fixed in these LLM model experiments. These tests were separate because `gpt-5` was not released until after the first test had already concluded. We report costs per model in Table 10.

**Case Study 2: Planning.** Given a potentially complex user message $M_i$, the agent typically takes multiple actions $A_i$ before asking for further feedback, thus potentially benefiting from having a plan of attack. In classical AI literature, planning has a long history of formalism and methods (Fikes & Nilsson, 1971; Blum & Furst, 1997; Kautz & Selman, 1992). However, with LLM-powered agents, recent approaches have largely adopted language reasoning as a medium for planning as opposed to symbolic operators and explicit transition models (Yao et al., 2022; Shinn et al., 2023). We investigate how showing agentic planning influences a user's experience. Specifically, the agent calls a `task_tracker` tool at the beginning of the conversation to create a structured task list, which is shown to the user as `TASKS.md` when encountering a complex, or multi-phase development task, to create a structured task list. For simple, atomic tasks, the agent will proceed with direct implementation to avoid tracking overhead. Agent updates task statuses as work progresses, and the frontend display is updated for the user accordingly.

**Case Study 3: Memory Management.** As the number of work segments increases in a given $X_i$, after a certain point, the entire history cannot fit into even state-of-the-art LLM context lengths. Additionally, it is no surprise that agents can become especially costly with long contexts (Anthropic, 2025b). There is a growing research community studying how to appropriately manage the context that is used as input to the agent (Jiang et al., 2023; Asai et al., 2024). In our implementation, as the conversation grows beyond a certain threshold, we intelligently summarize older interactions while keeping recent exchanges intact (Smith, 2025). This creates a concise "memory" of what happened earlier without needing to retain every detail. An important parameter in this setup is deciding *when* this thresholding occurs. We decrease `max_step` from 120 to 80, which leads to an expected amortized 0.5 cent savings *per step*, based on simulations on SWE-Bench (Jimenez et al., 2023a), which are described in Appendix A.3.

### 5.2. Analysis Procedure

While there was some variance across case studies, we collected at least 150 labels per condition for each case study and ran each comparison for 2-3 weeks. Randomization for A/B testing was conducted at the conversation level (represented as a trajectory), where each new conversation

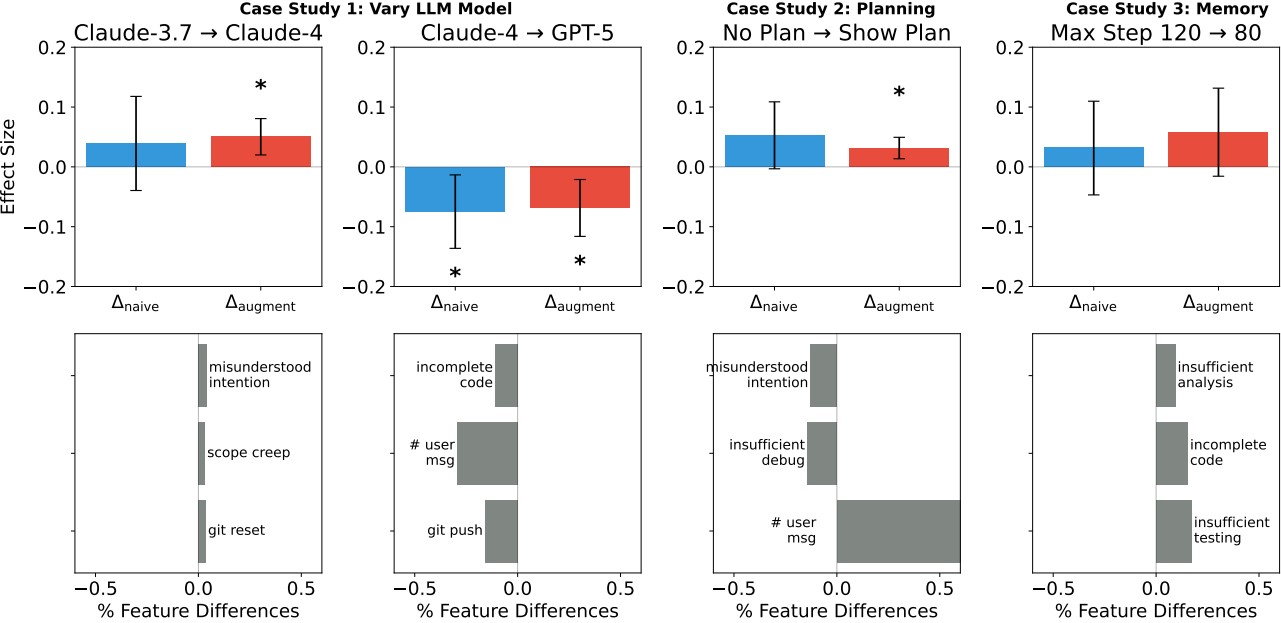

*Figure 4.* **How do user ratings change based on different software agent designs?** We report average user ratings (on a scale of 1-5) using only human labels and including `PULSE`. For fair comparison of the effect of PPI, we subsample to the same number of data points in human-only (i.e., 150 per condition). We use * to indicate significant results with a cutoff $\alpha = 0.05$. We can see that the LLM model makes the biggest difference (case study 1) compared to scaffold changes (case studies 2 and 3) and `PULSE` leads to more significant results (i.e., comparing $\Delta_{\text{naive}}$ and $\Delta_{\text{augment}}$).

was assigned a specific agent variant with fixed parameters. This design enables the same user to contribute multiple trajectories under different variants. If a user revisits a given conversation with the agent, the same test parameters would persist. For our analysis next, we use *the same feedback model*, rather than training a specific feedback model per case study. This is because the user population and bulk of the agent scaffold and user interaction process are fixed across case studies. To quantify the effect of agent design change in each case study, we report $\Delta_{\text{naive}}$ and $\Delta_{\text{augment}}$ using the best performing $f$ we trained (i.e., random forest model), along with the 95% confidence intervals. Additionally, we also compare conditions using features discussed in Section 3.2 to help interpret the observed effect sizes. In Figure 4 (bottom), we select the top 3 features based on p-value—note these are all trajectories that users rated, not with inferred ratings. We provide full result tables in Appendix B.

## 6. Results

### 6.1. Effect of agent designs on user satisfaction

**LLM backbone impacts user satisfaction more than scaffold changes.** Figure 4 (top) overviews the observed effect sizes across our case studies. Across both tests in case study 1 that vary the LLM models, users significantly preferred agents powered by `claude-4-sonnet` over the

two other LLMs. In the first test, we find a 5.86% difference in user satisfaction between `claude-3.7-sonnet` and `claude-4-sonnet`. In the second test, we find a -7.83% difference in user satisfaction between `claude-4-sonnet` and `gpt-5`. In contrast, for scaffolding changes, we find a small (but significant) 3.1% difference in user satisfaction between planning and no plan variants. Overall, the effect size is smaller than case study 1 where we change the LLM backbone, suggesting that the process is less important to the user (i.e., how the agent gets to a final goal) and rather the quality of the completed work.

**Interaction features provide further insight into how user experience changes.** Figure 4 (bottom) shows how features described in Section 3.2 across conditions change. We discuss some key observations: Since `gpt-5` was rated significantly lower than `claude-4-sonnet`, user interactions provide suggestions into why that might be. Trajectories with `gpt-5` contained 32% fewer user messages on average, which means users often stopped engaging and abandoned prompting earlier. We also observe 16% fewer code pushes when using `gpt-5`, suggesting that users decided it was not worth pushing incremental code edits when progress feels unproductive. In terms of the planning case study, we see that despite a small positive improvement in rating, there are multiple changes in behavioral features that indicate the benefits of showing plans to users. In particular, we see that in the no plan version, the agent is 12.8% more

*Table 2.* **Comparison of methods for predicting user satisfaction.** We evaluate two approaches: traditional ML models trained on labeled trajectories and LLM-as-a-judge (Zheng et al., 2023) across multiple long context models.

| Metric | LogReg | HGB | RF | LLM-as-a-Judge | | |
|---|---|---|---|---|---|---|
| | | | | o3 | gemini-2.5-pro | claude-4 |
| MSE ($\downarrow$) | $\mathbf{1.39}_{\pm 0.01}$ | $1.44_{\pm 0.02}$ | $1.44_{\pm 0.01}$ | $2.17_{\pm 0.01}$ | $2.52_{\pm 0.02}$ | $2.04_{\pm 0.03}$ |
| MAE ($\downarrow$) | $1.07_{\pm 0.01}$ | $1.03_{\pm 0.01}$ | $\mathbf{1.02}_{\pm 0.01}$ | $1.87_{\pm 0.02}$ | $2.05_{\pm 0.10}$ | $1.70_{\pm 0.04}$ |
| Correlation ($\uparrow$) | $0.24_{\pm 0.01}$ | $0.27_{\pm 0.02}$ | $\mathbf{0.29}_{\pm 0.01}$ | $0.22_{\pm 0.03}$ | $0.14_{\pm 0.07}$ | $0.23_{\pm 0.01}$ |

*Table 3.* **Human ratings do not always correlate with benchmarks.** We compare the difference observed in human ratings ($\Delta_H$) against the differences on 7 benchmarks ($\Delta_B$). We focus human ratings on the relevant subset most similar to the benchmark, ensuring each batch has at least 35 human data points.

| Task Type | Claude 3.7 vs Claude 4 | | Claude 4 vs GPT 5 | |
|---|---|---|---|---|
| | $\Delta_H$ | $\Delta_B$ | $\Delta_H$ | $\Delta_B$ |
| Testing code (Mündler et al., 2024) | 24.22% | 22.8% | 4.01% | 21.8% |
| Fix Continuous Integration (Bogomolov et al., 2024) | 20.04% | -4.35% | 24.05% | 28.87% |
| Fix Codebase Issues (Jimenez et al., 2023a) | 15.68% | 12.4% | 7.54% | 1.80% |
| Fix underspecified issues (Vijayvargiya et al., 2026) | 14.13% | 9.74% | -6.07% | 15.81% |
| Deep Research (Mialon et al., 2023) | 11.62% | 0.00% | -7.84% | 14.62% |
| Administrative tasks (Xu et al., 2024) | 4.05% | 2.28% | -4.62% | -0.75% |
| Write code from scratch (Zhao et al., 2024) | 0.64% | -5.50% | -17.9% | 19.0% |
| Pearson Correlation Coefficient ($\Delta_H, \Delta_B$) | $\rho =0.66$ | | $\rho =-0.18$ | |

likely to misunderstand the user, which leads to insufficient analysis and debugging (13.0% and 14.4% respectively). The increase in user messages also shows better engagement in agent work.

### 6.2. Measuring efficiency of agent evaluations

**PULSE can help provide more conclusive experimental results.** When comparing $\Delta_{\text{naive}}$ and $\Delta_{\text{augment}}$ across the 4 different experiments, we can see that confidence interval bands decrease on average by 39.5%. In fact, in multiple experiments—`claude-3.7-sonnet` versus `claude-4-sonnet`) and `plan` versus `no plan`, we find that we can draw more conclusive results that are statistically significant using $\Delta_{\text{augment}}$. More concretely, the 95% CI for $\Delta_{\text{naive}}$ in the `claude-3.7-sonnet` versus `claude-4-sonnet` comparison was , while the 95% CI of using augmented labels was [-3.95%, 11.78%], [1.99%, 8.06%]. The variation in CI reduction across case studies is largely due to how well $f$ can explain the specific samples for that particular test. Further, we observe that under realistic, noisy feedback conditions, we can use PULSE to detect significant differences across agent designs.

**Prediction models in PULSE outperform LLM-as-a-judge baselines.** Overall, we find that ML models trained on features extracted from interaction trajectories significantly outperform state-of-the-art LLMs across all metrics considered (Table 2). In particular, predictive methods im-

prove correlation with outcomes by at least 26% compared to baseline. Across all three LLMs evaluated, we find that LLMs tend to be more pessimistic than users. For example, o3 rarely gives a score of 5. An additional benefit of training a model with interpretable features is that we can use it to understand what aspects of agent behavior lead to more or less user satisfaction. Across models, we find the most important features to be `user sentiment` based on messages and `git push` based on agent actions—the importance of task completion features shows that users care not just about interaction with the agent but about task completion. However, no feature alone is fully predictive of user rating.

## 7. Correlating results with benchmarks

Finally, we present an exploratory analysis of how our findings compare to benchmarks. In particular, we focus on our first case study on varying LLM models because that led to the most drastic changes in human ratings. We consider a variety of 7 benchmarks that test agentic software engineering tasks that range from improving code bases (Jimenez et al., 2023b; Vijayvargiya et al., 2026) to writing tests and fixing continuous integration (Mündler et al., 2024; Bogomolov et al., 2024). We categorize human-agent trajectories into corresponding categories of tasks and compare $\Delta_H$ (i.e., the change in human ratings) and $\Delta_B$ (i.e., the change in benchmark performance).

**Static benchmarks don't tell the whole story.** Table 3 shows the comparison between $\Delta_H$ and $\Delta_B$ on the case study that varies LLM models. We find a moderate positive correlation on the `claude-3.7-sonnet` and `claude-4-sonnet` comparison. However, we observe a weak negative correlation on the `claude-4-sonnet` and `gpt-5` (i.e., $\rho = 0.66$ vs $-0.11$). This means that we should not always take benchmark improvements at face value as there may be additional deployment challenges when humans are involved. Interestingly, we see the most alignment in tasks like testing (e.g., SWT-Bench (Mündler et al., 2024)) and administrative tasks (e.g., The Agent Company (Xu et al., 2024))—where both $\Delta_H$ and $\Delta_B$ are the same sign—which are not the standard type of "bug fixing" task that many agentic coding benchmarks are built around. Similarly, we find the largest $|\Delta_H|$ to be from fixing continuous integration issues, rather than simply working on the code base. These results suggest the importance of moving beyond SWE-Bench-like evaluations even for benchmarks.

## 8. Discussion and Conclusion

As humans increasingly collaborate with AI agents in real-world settings, evaluation protocols must reflect these interaction dynamics rather than assuming full automation. In this work, we introduced `PULSE`, a framework for efficiently measuring the effects of agent design choices in human–agent interactions, and deployed it across multiple large-scale case studies of software engineering agents. Our results underscore the importance of human-in-the-loop evaluation and show that benchmark performance does not always align with user experience. As LLM-powered applications continue to see rapid deployment across academia and industry, we see substantial opportunity for applying our framework beyond software engineering to other domains where agents and humans work collaboratively.

**Limitations and Future Work.** Although our study spans a diverse set of users and real-world tasks, our case studies focus on a single agent platform (OpenHands). Future work should apply `PULSE` to additional agents and domains to better understand how design insights transfer. We also use user ratings as our primary human-centered metric, but do not explicitly model label noise. Future work should explore richer measures of agent performance, as well as statistical approaches for noise-aware inference. Finally, due to privacy constraints, we cannot release raw code contexts collected during the study. To support reproducibility, we will release the `PULSE` framework code and extensions to the OpenHands platform, along with anonymized feature-level datasets.

## Impact Statement

This work advances methods for evaluating human–AI collaboration in real-world settings. By improving how agent systems are assessed with human feedback, our framework may influence how AI tools are developed and deployed in professional environments such as software engineering. More reliable evaluation of human–agent interaction could help align AI systems with user needs, potentially improving productivity, usability, and trust.

At the same time, methods that enable more effective AI agents may contribute to increased automation of knowledge work, which can have complex implications for labor, job roles, and skill requirements. Future work should continue to consider how evaluation practices shape the incentives and real-world impact of AI deployment.

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

# A. Additional Framework Details

## A.1. Data collection details

To make feedback minimally intrusive, we prompt users at the end of each work segment to rate the agent's performance on a five-star scale as shown in Figure 5.

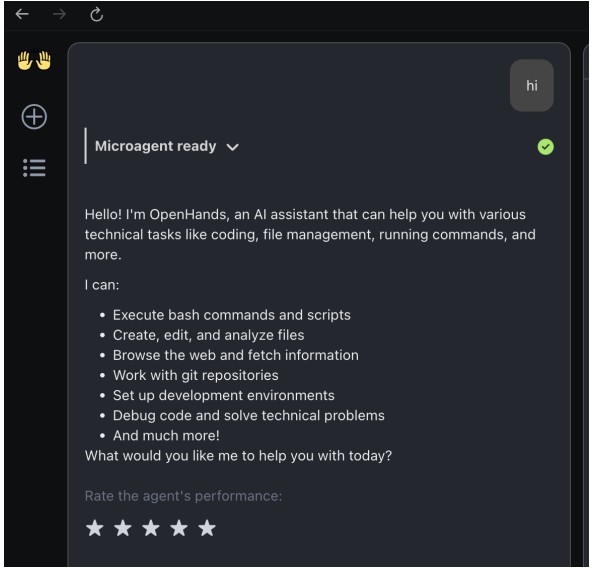

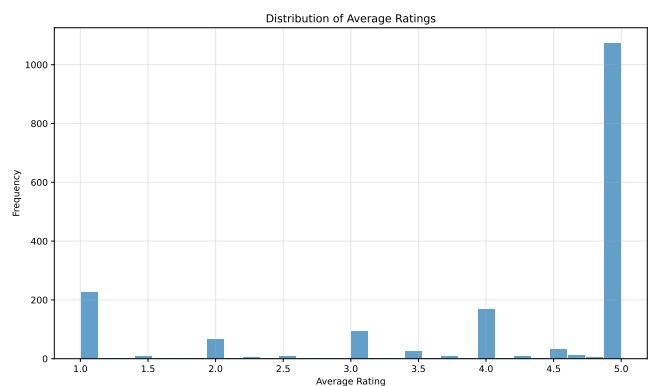

*(a)* Toy example showing the 5-star feedback interface.        *(b)* Distribution of ratings.

*Figure 5.* Illustration of the feedback setup: (a) the 5-star feedback interface, and (b) the distribution of ratings collected.

## A.2. Predicting user satisfaction

**Feature list.** We identified 15 features used to train ML model to predict user satisfaction ratings.

1. User sentiment: positive, negative, neutral

2. Number of user messages

3. Task Category: we identified 8 common tasks that developers use software agents for.

4. Misunderstood Intention: Agent misunderstood the user's goal or intent.

5. Did not follow instruction: Agent ignored or failed to comply with explicit

6. Insufficient Analysis: Didn't explore existing materials sufficiently (prior code/docs/examples) before acting.

7. Insufficient testing: Skipped reasonable verification/tests for non-trivial or risky changes (note: trivial edits may be acceptable).

8. Insufficient Debugging: Did not investigate or reduce failing behavior when needed to make progress.

9. Incomplete Implementation: Delivered unfinished or non-functioning work.

10. Scope Creep: Implemented unrequested features without approval.

11. `git_commit`

12. `git_push`

13. `git_pull`

14. `git_reset`

15. `git_rebase`

We use LLM as a judge to detect features 1,3,4,5,6,7,8,9,10. We analyze the event stream to detect features 2,11,12,13,14,15. Below we include the prompt used with `gpt-5-mini`.

**Prompt for labeling features**

```
Analyze the following user messages from a coding assistant session:

{combined_messages}

Please provide the following analysis:
1. One sentence describing what the user is trying to accomplish
2. Classify the overall sentiment of the user's messages into one of these
   categories: [Positive, Negative, Neutral] and explain why
3. Classify the type of task into exactly one of these categories (choose only
   one that best fits): [Fix Bugs, Implement Features, Create Programs from
   Scratch, Fix Failing Continuous Integration, Fix Merge Conflicts, Write
   Documentation, Perform Deployments, Perform Data Analysis]
4. Classify the development cluster into up to two of these categories (choose
   only the ones that are the best fits):
   - Write code from scratch
   - Fix python issues
   - Fix underspecified issues
   - Fix Java issues
   - Testing code
   - Web browsing and research
   - Administrative tasks
   - Fix continuous integration issues
   - None of the above
5. Provide 1-2 brief example messages from the conversation that support your
   sentiment classification (truncate if too long)

Format your response as JSON with these fields:
{{
   "task_description": "one sentence",
   "sentiment": {{
       "classification": "Positive/Negative/Neutral",
       "explanation": "brief explanation",
       "example_messages": ["message 1", "message 2"]
   }},
   "task_type": "one of the categories",
   "development_cluster": ["cluster 1", "cluster 2" (if applicable)]
}}
```

```
Analyze the following user messages from a coding assistant session:

{combined_messages}

For each issue item below, answer YES or NO based on whether there is evidence in
    the user messages that this issue occurred. Provide a brief explanation for
    each:

    [item 1] misunderstood_intention: Agent misunderstood the user's goal/intent.
    - Examples: User asked for a summary and agent produced a rewrite; user wanted
        high-level bullets but agent delivered full code.

    [item 2] did_not_follow_instruction: Agent ignored or failed to comply with
        explicit instructions/system constraints.
    - Examples: User: "Do NOT push to main." Agent pushes to main; System says not
        to create pull request unless user asks for it and user didn't ask for it
        , agent creates pull request; user asked for bullet points only, agent
        gives long prose.

    [item 3] insufficient_analysis: Didn't explore existing materials sufficiently
        (prior code/docs/examples) before acting.
    - Examples: User points to an existing function/file that is relevant OR
        already solves it; agent reinvents it.

    [item 4] insufficient_testing: Skipped reasonable verification/tests for non-
        trivial or risky changes (note: trivial edits may be acceptable).
    - Examples: No run/validation for a new parser; no check that a migration
        applies cleanly; no sanity check of output.

    [item 5] insufficient_debugging: Did not investigate or reduce failing
        behavior when needed to make progress.
    - Examples: Ignores stack trace; no isolation of failure; proceeds while
        errors persist.

    [item 6] incomplete_implementation: Delivered unfinished or non-functioning
        work.
    - Examples: TODO/FIXME left; stub methods; code that cannot run.

    [item 7] scope_creep: Implemented unrequested features without approval.
    - Examples: Adds a dashboard or endpoint not asked for.

Format your response as JSON with these fields:
```

**Exploring Automatic Discovery of Features.** We create a script that takes as input a user-agent interaction trajectory and queries one or more LLMs to propose 10 general, reusable features that predict user satisfaction, using each conversation as contextual examples. The script calls models asynchronously and creates a JSONL of per-conversation/per-model outputs plus an aggregated per-model summary of recurring features. The authors of the paper then aggregate features generated by each model and compare them to the ones used in our study. Table 4 shows the results across 5 models.

*Table 4.* With a simple prompting scheme, we find that multiple models, including claude-opus-4.5 and claude-4-sonnet can re-identify a majority of our features.

| Model | Feat about user | Feat about agent | Feat about task progression | Overall (↑) |
|---|---|---|---|---|
| o3 | 2/2 | 5/8 | 0/5 | 7/15 |
| gpt-4o | 2/2 | 4/8 | 2/5 | 8/15 |
| gpt-5.1 | 2/2 | 7/8 | 0/5 | 9/15 |
| claude-opus-4.5 | 2/2 | 7/8 | 2/5 | 12/15 |
| claude-4-sonnet | 2/2 | 6/8 | 3/5 | 12/15 |

**Prompt for identifying features**

```
You are an expert product analyst. Using the following conversation ONLY as an
    example of the kinds of signals available (user/agent messages, visible
    errors, git actions, etc.), propose 10 general, reusable features that would
    predict user satisfaction or dissatisfaction across coding-assistant
    conversations.

Do NOT evaluate this particular conversation. Define features that apply broadly.
     Each feature should be clearly computable from typical traces like this one.

EXAMPLE CONVERSATION CONTEXT (for reference only)
---
{conversation}
---

Return ONLY valid JSON with this schema:
{{
  "features": [
    {{
      "name": "short, canonical feature name",
      "definition": "what the feature measures in general terms",
      "why_predictive": "why the feature correlates with satisfaction/
          dissatisfaction",
      "how_to_compute": "signals/heuristics to extract it from conversation logs"
          ,
      "value_type": "one of: binary | ordinal | count | ratio | continuous",
      "polarity": "satisfaction | dissatisfaction | mixed",
      "example_indicators": ["short, concrete examples of signals"]
    }},
  ],
  "overall_guidance": "one sentence on how to use this feature set"
}}

Constraints:
- Provide exactly 10 features
- Do not include per-conversation judgments or values
- Keep definitions concise and implementation-oriented
- Respond with JSON only (no prose outside JSON)
```

**ML Model set-up and parameters:** We consider the following model families

- Ordinal logistic "regressor" (Pipeline: DictVectorizer $\rightarrow$ to_dense $\rightarrow$ StandardScaler(with_mean=True) $\rightarrow$ OrdinalLogitRegressor(C=2.0, class_weight='balanced'))

- Random forest regressor (Pipeline: DictVectorizer $\rightarrow$ to_dense $\rightarrow$ RandomForestRegressor(n_estimators=400))

- HistGradientBoostingRegressor (Pipeline: DictVectorizer $\rightarrow$ to_dense $\rightarrow$ HistGradientBoostingRegressor())

We perform k-fold cross-validation with 5 splits with a fixed random seed (42). For each fold, we fit the model on train indices, predict on test indices, collect out-of-fold predictions, then finally computes metrics on all out-of-fold predictions combined.

To reduce the effect of outliers, for models such as logistic regression, we explicitly include regularization to reduce sensitivity to outliers, while Random Forests and Histogram Gradient Boosting already incorporate strong forms of implicit regularization.

**Prompt for LLM-as-a-judge baseline**

```
Please analyze the following conversation between a user and an AI coding
    assistant to determine user satisfaction.

CONVERSATION:
{conversation}

Please provide your analysis in the following JSON format:
{{
    "binary_satisfied": true/false,
    "likert_score": 1-5,
    "explanation": "detailed explanation of your reasoning"
}}

EVALUATION CRITERIA:
- Binary satisfaction: Was the user satisfied with the agent's help by the end of
    the conversation? (true/false)
- Likert scale (1-5): 1=Very Dissatisfied, 2=Dissatisfied, 3=Neutral, 4=Satisfied
    , 5=Very Satisfied
- Consider factors like:
  - Whether the user's problem was solved
  - Quality of the agent's responses
  - User's tone and feedback throughout the conversation
  - Whether the user expressed gratitude or frustration
  - If the conversation ended on a positive or negative note

Respond ONLY with the JSON format above, no additional text.
```

### A.3. Selecting Memory Management Parameters

To select the appropriate `max_step` parameters for case study 3 we ran OpenHands on a randomly-selected subset of 50 problems from the Verified subset of SWE-Bench while varying `max_step` from 20 to 160 in increments of 20. OpenHands was configured to run at most 200 steps, but trajectory lengths varied based on the problem instance. For each value of `max_step` we averaged the API cost per-step across all problem instances and applied locally-weighted scatterplot smoothing with a bandwidth of 0.1 to better illustrate the amortized cost behavior. The results are given in Figure 6.

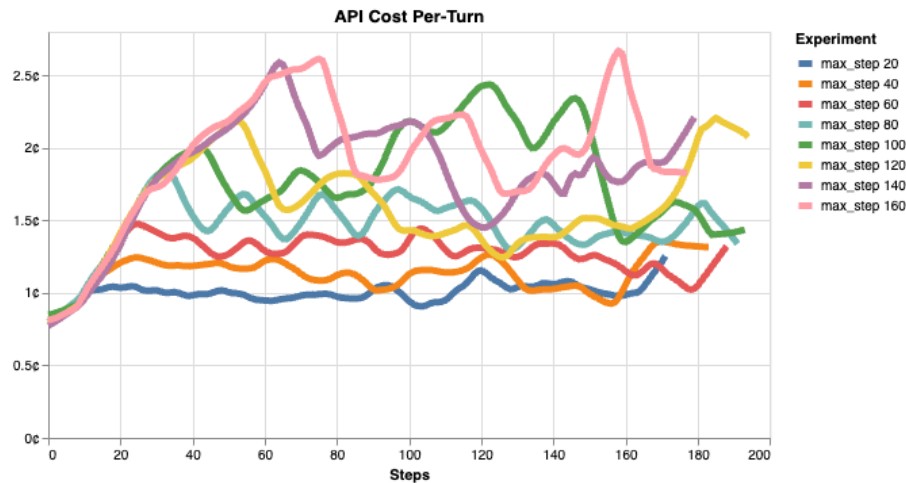

*Figure 6.* Average API cost per-step with varying `max_step` values.

To ensure that changing `max_step` did not meaningfully impact agent performance we evaluated the trajectories associated with the selected values (80 and 120) using the SWE-bench evaluation framework. The framework was able to evaluate trajectories for 22 of the 50 problem instances: with `max_step` set to 80, OpenHands resolved 16 problems, and with `max_step` set to 120 OpenHands resolved 15 problems.

## B. Full Results

A full table of effect sizes across all experiments is provided in Table 5. We provide full feature comparison for each case study: Claude 4 vs. GPT-5 (Table 6), Claude 3.7 vs. Claude 4 (Table 7), Condenser 120 vs. 80 (Table 8), and Planning vs. No-planning (Table 9). We also present average costs per model in Table 10. Finally, we also show the relative reduction in confidence intervals using other model families in Table 11.

*Table 5.* Effect sizes ($\Delta$) with 95% confidence intervals. An asterik (*) indicates the CI does not include 0.

| Comparison | Condition | Effect | CI Lower | CI Upper | Sig. |
|---|---|---|---|---|---|
| Claude-3.7 → Claude-4 | $\Delta_{\text{naive}}$ | 0.039 | -0.040 | 0.118 | |
| | $\Delta_{\text{augment}}$ | 0.050 | 0.020 | 0.081 | * |
| Claude-4 → GPT-5 | $\Delta_{\text{naive}}$ | -0.075 | -0.136 | -0.013 | * |
| | $\Delta_{\text{augment}}$ | -0.069 | -0.116 | -0.021 | * |
| No Plan → Show Plan | $\Delta_{\text{naive}}$ | 0.053 | -0.003 | 0.109 | |
| | $\Delta_{\text{augment}}$ | 0.031 | 0.014 | 0.049 | * |
| Memory max step | $\Delta_{\text{naive}}$ | 0.032 | -0.047 | 0.110 | |
| | $\Delta_{\text{augment}}$ | 0.058 | -0.016 | 0.132 | |

## C. Opportunities for Agent Design

Our case studies show that user satisfaction is more sensitive to the choice of LLM backbone than to scaffolding changes, with `claude-4-sonnet` consistently outperforming alternatives. Interaction features also provide additional context, revealing patterns such as early disengagement and reduced code pushes with `gpt-5`, and improved engagement when plans are surfaced to users. We identify two directions for future work and show how our framework can be adapted to other domains in Table 12:

**More Interactive LLM Backbones.** Since our results suggest that improvements in backbone quality remain the primary driver of user satisfaction, it also highlights an opportunity to train backbones that are tuned for interactivity. Rather than optimizing solely for benchmark accuracy, future models could emphasize capabilities that users find desirable in collaborative contexts. For example, this might include maintaining a consistent multi-turn state, dynamically clarifying ambiguous user intent, and proactively getting feedback. Such properties would support more sustained engagement and reduce failure modes like misunderstandings or abandoned sessions, as we observed in our study.

**Better Modeling of User Satisfaction and Engagement.** Our findings also underscore the value of modeling user satisfaction. We find signals like early disengagement, frequency of corrective messages, or code push behavior can serve as early indicators of dissatisfaction, which can complement sparse, explicit rating data. We encourage future work to explore how these behavioral features in human-agent interactions can be studied in more detail to see if they can be used to learn adaptive or more personalized interventions in real time.

## D. Disclosure

The authors used ChatGPT for minor copyediting tasks.

*Table 6.* Claude-4 → GPT-5 Feature Comparison.

| Feature | Var A | Var B | Mean A | Mean B | Diff (B-A) | p-val |
|---|---|---|---|---|---|---|
| Misunderstood intention | control | gpt5 | 0.2771 | 0.1805 | -0.097 | 0.039 |
| Did not follow instruction | control | gpt5 | 0.3463 | 0.2857 | -0.061 | 0.235 |
| Insufficient analysis | control | gpt5 | 0.3723 | 0.2857 | -0.087 | 0.094 |
| Insufficient testing | control | gpt5 | 0.3550 | 0.3008 | -0.054 | 0.292 |
| Insufficient debugging | control | gpt5 | 0.3723 | 0.2932 | -0.079 | 0.127 |
| Incomplete implementation | control | gpt5 | 0.4589 | 0.3459 | -0.113 | 0.036 |
| Scope creep | control | gpt5 | 0.1212 | 0.0902 | -0.031 | 0.364 |
| User message count | control | gpt5 | 19.17 | 13.52 | -5.65 | 0.028 |
| Git commit | control | gpt5 | 0.6580 | 0.6241 | -0.034 | 0.515 |
| Git push | control | gpt5 | 0.6104 | 0.4511 | -0.159 | 0.003 |
| Git pull | control | gpt5 | 0.0866 | 0.0827 | -0.004 | 0.900 |
| Git reset | control | gpt5 | 0.0909 | 0.0526 | -0.038 | 0.188 |
| Git rebase | control | gpt5 | 0.0173 | 0.0150 | -0.002 | 0.871 |

*Table 7.* Claude-3.7 → Claude-4 Feature Comparison

| Feature | Var A | Var B | Mean A | Mean B | Diff (B-A) | p-val |
|---|---|---|---|---|---|---|
| Misunderstood intention | Claude 4 | Claude 3.7 | 0.2900 | 0.2491 | -0.041 | 0.105 |
| Did not follow instruction | Claude 4 | Claude 3.7 | 0.3940 | 0.3764 | -0.018 | 0.524 |
| Insufficient analysis | Claude 4 | Claude 3.7 | 0.4309 | 0.4244 | -0.007 | 0.815 |
| Insufficient testing | Claude 4 | Claude 3.7 | 0.4104 | 0.3911 | -0.019 | 0.489 |
| Insufficient debugging | Claude 4 | Claude 3.7 | 0.4186 | 0.4336 | 0.015 | 0.593 |
| Incomplete implementation | Claude 4 | Claude 3.7 | 0.4952 | 0.4797 | -0.016 | 0.584 |
| Scope creep | Claude 4 | Claude 3.7 | 0.1231 | 0.0904 | -0.033 | 0.064 |
| User message count | Claude 4 | Claude 3.7 | 16.41 | 15.77 | -0.64 | 0.641 |
| Git commit | Claude 4 | Claude 3.7 | 0.5335 | 0.5277 | -0.006 | 0.837 |
| Git push | Claude 4 | Claude 3.7 | 0.4829 | 0.5018 | 0.019 | 0.504 |
| Git pull | Claude 4 | Claude 3.7 | 0.0602 | 0.0849 | 0.025 | 0.090 |
| Git reset | Claude 4 | Claude 3.7 | 0.0848 | 0.0480 | -0.037 | 0.010 |
| Git rebase | Claude 4 | Claude 3.7 | 0.0205 | 0.0111 | -0.009 | 0.191 |

*Table 8.* Memory max step treatment (120) and Control (80) Feature Comparison

| Feature | Var A | Var B | Mean A | Mean B | Diff (B-A) | p-val |
|---|---|---|---|---|---|---|
| Misunderstood intention | Treatment | Control | 0.2308 | 0.2329 | 0.002 | 0.981 |
| Did not follow instruction | Treatment | Control | 0.3846 | 0.3014 | -0.083 | 0.335 |
| Insufficient analysis | Treatment | Control | 0.4231 | 0.3288 | -0.094 | 0.284 |
| Insufficient testing | Treatment | Control | 0.4615 | 0.2877 | -0.174 | 0.047 |
| Insufficient debugging | Treatment | Control | 0.4615 | 0.3699 | -0.092 | 0.307 |
| Incomplete implementation | Treatment | Control | 0.5385 | 0.3836 | -0.155 | 0.088 |
| Scope creep | Treatment | Control | 0.0769 | 0.0411 | -0.036 | 0.396 |
| User message count | Treatment | Control | 19.08 | 18.03 | -1.05 | 0.307 |
| Git commit | Treatment | Control | 0.5577 | 0.5479 | -0.010 | 0.917 |
| Git push | Treatment | Control | 0.5192 | 0.4932 | -0.026 | 0.777 |
| Git pull | Treatment | Control | 0.0769 | 0.0685 | -0.008 | 0.862 |
| Git reset | Treatment | Control | 0.0577 | 0.0548 | -0.003 | 0.950 |
| Git rebase | Treatment | Control | 0.0000 | 0.0274 | 0.027 | 0.235 |

*Table 9.* No Plan → Show Plan Feature Comparison.

| Feature | Var A | Var B | Mean A | Mean B | Diff (B-A) | p-val |
|---|---|---|---|---|---|---|
| Misunderstood intention | Planning | Control | 0.2222 | 0.2771 | 0.055 | 0.168 |
| Did not follow instruction | Planning | Control | 0.3128 | 0.3463 | 0.034 | 0.438 |
| Insufficient analysis | Planning | Control | 0.3004 | 0.3723 | 0.072 | 0.098 |
| Insufficient testing | Planning | Control | 0.2963 | 0.3550 | 0.059 | 0.173 |
| Insufficient debugging | Planning | Control | 0.2922 | 0.3723 | 0.080 | 0.064 |
| Incomplete implementation | Planning | Control | 0.3868 | 0.4589 | 0.072 | 0.113 |
| Scope creep | Planning | Control | 0.0988 | 0.1212 | 0.022 | 0.435 |
| User message count | Planning | Control | 13.16 | 19.17 | 6.01 | 0.070 |
| Git commit | Planning | Control | 0.6461 | 0.6580 | 0.012 | 0.786 |
| Git push | Planning | Control | 0.5514 | 0.6104 | 0.059 | 0.194 |
| Git pull | Planning | Control | 0.0947 | 0.0866 | -0.008 | 0.761 |
| Git reset | Planning | Control | 0.0782 | 0.0909 | 0.013 | 0.619 |
| Git rebase | Planning | Control | 0.0165 | 0.0173 | 0.001 | 0.944 |

*Table 10.* Cost comparison across LLMs.

| Model | Avg Cost and Std Dev Per Conversation |
|---|---|
| *Claude-3.7-Sonnet vs Claude-4-Sonnet* | |
| Claude-3.7-Sonnet | $2.62 \pm $3.80 |
| Claude-4-Sonnet | $2.63 \pm $4.22 |
| *Claude-4-Sonnet vs GPT-5* | |
| Claude-4-Sonnet | $3.05 \pm $5.90 |
| GPT-5 | $2.97 \pm $6.50 |

*Table 11.* Percent CI Reduction (higher is better)

| | **LogReg** (*Corr = 0.24* ) | **HGB** ( *Corr = 0.27*) | **RF** (*Corr = 0.29*) |
|---|---|---|---|
| Claude-3.7 vs Claude-4 | 3.57 | 17.55 | 61.5 |
| Claude-4-Sonnet vs GPT-5 | 5.5 | 6.61 | 26 |
| Planning vs no planning | 4.23 | 50.93 | 65 |
| Memory Max Step | 2.8 | 0.96 | 5.5 |

*Table 12.* Overview of how future work can apply our three-step framework to varying domains to evaluate human-agent interaction: shopping with web agent, conducting experiments and literature search with research agent, and planning trip with travel agent.

| | **Shopping Agent** | **Research Agent** | **Trip Planning Agent** |
|---|---|---|---|
| **Step 1:** | Create an interface that periodically collects 5-star ratings as the agent is adding items to the cart. | Create an interface that periodically collects 5-star ratings as the agent is collecting literature and running experiments. | Create an interface that periodically collects 5-star ratings as the agent builds a travel itinerary. |
| **Step 2:** | Train ML model using user features (number of messages and sentiment), agent features (shopping site, item category), and task progression (whether a purchase was made). | Train ML model using user features (number of messages and sentiment), agent features (research topic), and task progression (success of experiments). | Train ML model using user features (number of messages and sentiment), agent features (travel type), and task progression (whether a hotel was reserved). |
| **Step 3:** | Apply directly: the methodology to compare $\widehat{\Delta}_{\text{naive}}$ and $\widehat{\Delta}_{\text{augment}}$ is entirely agnostic to the software engineering use case. | Apply directly: the methodology to compare $\widehat{\Delta}_{\text{naive}}$ and $\widehat{\Delta}_{\text{augment}}$ is entirely agnostic to the software engineering use case. | Apply directly: the methodology to compare $\widehat{\Delta}_{\text{naive}}$ and $\widehat{\Delta}_{\text{augment}}$ is entirely agnostic to the software engineering use case. |

