# OpenReview forum: "How can we assess human-agent interactions? Case studies in software agent design"
_ICML.cc/2026/Conference — ICML 2026 regular_

### Official Review · Reviewer_quto · 2026-03-08

**Soundness:** 3
**Presentation:** 3
**Significance:** 4
**Originality:** 2
**Overall Recommendation:** 5
**Confidence:** 4

**Summary:**

This paper introduces the PULSE framework to evaluate coding agent A/B tests (varying both underlying LLMs and agent scaffolding) under conditions of limited user feedback. Furthermore, it reveals significant discrepancies between static benchmark results and real-world human evaluations, highlighting the critical need for reliable, human-in-the-loop assessment protocols in the agent domain.

**Compliance With Llm Reviewing Policy:**

Affirmed.

**Final Justification:**

The authors have addressed my concern, and I will raise my score to 5.

**Key Questions For Authors:**

Please see the weakness above.

**Limitations:**

yes

**Strengths And Weaknesses:**

Strengths:
- The paper addresses a critical and timely issue in the LLM agent domain, recognising that static benchmarks are no longer sufficient and have become disconnected from in-the-wild applications. Furthermore, the authors choose a highly practical entry point by tackling the reality that real human feedback is inherently sparse.
- The assessment is solidly grounded in a large-scale, real-world deployment with actual users, which adds substantial empirical value to the study.
- The authors present statistically significant findings that effectively expose the limitations and flaws of relying solely on static benchmarks.

Weaknesses:
- The application of PPI fundamentally relies on the flawed assumption that labelled and unlabeled trajectories share the same underlying distribution. In reality, severe self-selection bias exists, as users who voluntarily submit ratings, such as power users or those experiencing extreme satisfaction or frustration, systematically differ from the silent majority. A
- The reliance on 15 manually defined features feels naive and overly engineered. Many of these inputs, such as "misunderstood intention" or "did not follow instruction," are highly abstracted, LLM-generated evaluations that already serve as strong standalone metrics. Using a traditional machine learning model to aggregate these near-linear features into a single score is redundant, and any upstream extraction errors made by the LLM will heavily distort the final outcome.
- The overall framework lacks significant methodological novelty. It essentially ports a standard pipeline of collecting user feedback and training a satisfaction scoring model from traditional chatbot domains directly into the LLM agent space.
- The methodology resembles manual feature engineering for a reward model, overlooking established RLHF paradigms. Instead of aggregating features with traditional ML, fine-tuning an LLM to simulate user ratings could leverage strong priors to learn generalizable structures directly from sparse feedback. This approach would effectively bypass the need to explicitly debias massive amounts of unlabeled data.

---

> ### Author Rebuttal · Authors · 2026-03-30
>
> We thank the reviewer for recognizing that our work addresses a “critical and timely issue in the LLM agent domain” and for highlighting the value of our “solidly grounded in a large-scale, real-world deployment with actual users, which adds substantial empirical value to the study.”
>
> **W1: The application of PPI fundamentally relies on the flawed assumption that labelled and unlabeled trajectories share the same underlying distribution.**
>
> Self-selection bias would be problematic if our goal were to estimate the absolute satisfaction of a specific agent design. However, PULSE is used for comparative evaluation of agent variants. In this case, we argue that selection bias is comparable across variants because when someone chooses to provide feedback is likely independent of agent variants (e.g., claude-3.7-sonnet vs. claude-4-sonnet).
>
> Further, we empirically compared labeled and unlabeled trajectories across all 15 features used in our predictive model and quantified differences using rank-biserial correlation. Overall, we found that only the user message count exhibits a moderate effect, with labeled trajectories containing more user messages (RBC=0.32). All remaining differences between labeled and unlabeled trajectories are small or negligible (RBC around 0.1 or less). The primary difference in user message count suggests that explicit feedback tends to be provided by more verbose or engaged users, but unlabeled trajectories still exhibit similar patterns of agent failures and user sentiment, indicating that silent users likely experience similar issues but may not choose to provide explicit ratings.
>
> Finally, the presence of power users is an unlikely explanation as only 12.75% of users contributed ratings across multiple sessions, and among these users, the majority contributed only two sessions.
>
> In the final version, we will make sure to emphasize the conceptual differences in Section 3.3 and the observed empirical results in Section 4.1.
>
> **W2: The reliance on 15 manually defined features feels naive and overly engineered. Using a traditional machine learning model to aggregate these near-linear features into a single score is redundant.**
>
> Regarding “manually defined features,” we note that automated approaches can recover the majority of these features by prompting LLMs with few-shot trajectories (L216). However, the same models are unable to adequately use these features to predict human judgments, as our LLM-as-a-judge baseline in Table 2 shows systematic differences between LLM judgments and user ratings (i.e., across all three models evaluated, LLMs tend to be more pessimistic than users). This suggests that it is not redundant to train a separate model to aggregate features.
>
> Additionally, since PPI reduces variance when predictions are correlated with the true outcomes, we also show in Table 2 that nonlinear models outperform linear ones in terms of correlation with user ratings.
>
> **W3: The overall framework lacks significant methodological novelty.**
>
> We refer to ICML reviewing guidelines about “novelty”, which states that “originality does not necessarily require introducing an entirely new method. Rather, a work that provides novel insights by evaluating existing methods, or demonstrates improved understanding is also equally valuable.” We believe we have provided novel insights based on the reviewer’s noted strengths: “present statistically significant findings that effectively expose the limitations and flaws of relying solely on static benchmarks.”
>
> **W4: Instead of aggregating features with traditional ML, fine-tune an LLM to simulate user ratings. This approach would effectively bypass the need to explicitly debias massive amounts of unlabeled data.**
>
> We agree that fine-tuning an LLM to simulate user ratings is an interesting direction, and we encourage future work to explore such approaches, particularly in settings with a larger amount/percentage of explicit feedback. However, regardless of whether predictions are generated by a traditional ML model or a fine-tuned LLM, the pipeline for combining labeled and unlabeled trajectories would be the same. In the final version, we will also expand our related work to cover RLHF and other paradigms of learning from human feedback.

---

> > ### Author Rebuttal · Reviewer_quto · 2026-04-02
> >
> > The authors have addressed my concern, and I will raise my score to 5.

---

### Official Review · Reviewer_r5Po · 2026-03-12

**Soundness:** 2
**Presentation:** 2
**Significance:** 3
**Originality:** 4
**Overall Recommendation:** 5
**Confidence:** 3

**Summary:**

This paper argues that standard LLM-agent benchmarks fail to capture real-world human–agent collaboration due to its dynamic nature and proposes PULSE to evaluate agent design changes using user ratings plus model-generated pseudo-labels and prediction-powered inference. They claim model choice changes satisfaction by ~6–8% while scaffolding changes are smaller (<3%), that PULSE reduces CI widths ~40% vs. naive A/B, and that benchmark deltas correlate poorly with in-the-wild ratings.

**Compliance With Llm Reviewing Policy:**

Affirmed.

**Final Justification:**

I thank the authors for their detailed rebuttal, which has satisfactorily addressed my primary methodological and presentation concerns. My initial assessment recognized the paper's significant strengths: its large-scale in-the-wild evaluation, its focus on a genuine problem in human-agent collaboration, and its provision of a valuable dataset and framework.

The main weaknesses pertained to soundness and clarity, and the authors' responses have effectively mitigated these concerns, or they agreed to clarify them in the revised manuscript.

### 1. Soundness: This was my most substantial area of concern. The authors provided direct, data-driven answers to each point: a) Feature extraction & quantitative gaps, b) Selection Bias (5% feedback rate), 3. Nested Data Structure, 4) Causal Claim
### 2. Originality & Significance
### 3. Clarity & Presentation

The work provides a valuable framework (PULSE), a unique large-scale dataset, and compelling evidence that will influence how the community evaluates interactive AI agents. The authors' rebuttal has resolved my key concerns regarding methodological soundness and presentation. I have therefore upgraded my assessment and recommend acceptance.

**Key Questions For Authors:**

1. Which specific features does the ML satisfaction model extract?
2. What proportion of users participated in multiple sessions, and how does clustering affect results?
3. How do you address the 5% feedback selection bias?

**Limitations:**

yes

**Strengths And Weaknesses:**

Soundness of Technical Claims and Methodology
Plus :
-The authors present a large-scale deployment (15k users), which is rare in academia and strengthens empirical validity
-The authors use of human feedback implementation after each command is a well-designed methodological choice
-the authors perform benchmark-vs-human comparison provides valuable measurement contribution

Minus:
-However, the claim "We train an ML model to predict user satisfaction by extracting important features" lacks critical details, such as which features were extracted.
-Line 151 states LLM-based approaches "struggle" and predictive methods "outperform baselines", but by how much? What specific problems do dialogue-specialized LLMs encounter? Here, the reader would benefit from quantitative details
-Most importantly, only 5% of sessions received feedback, likely skewing toward users with stronger opinions. This could artificially reduce variance in satisfaction scores (AI ratings)
- That users participated multiple times creates dependency and a nested data structure. Can the authors provide any information on how many users contributed multiple sessions? Moreover, cluster-robust methods should be employed to account for this
-The observation that GPT-5 was rated significantly lower than Claude-4-sonnet is presented as a causal explanation for user behavior differences, which is a bold claim without controlled experimental evidence

Presentation and Contextualization
minus:
-The contribution section conflates technical/theoretical contributions (PULSE framework) with practical implementation details (experimental design, model selection, scaffolding). To support understanding of your theoretical foundation, these should be clearly differentiated
-Feature contribution analysis appears before PULSE framework evaluation. I would suggest considering restructuring to lead with the primary contribution
-Overall clarity would benefit from separating methodological innovations from empirical findings

Significance of Contribution
plus:
-The authors address a genuinely important problem of evaluating agents in realistic human-agent collaboration settings rather than static benchmarks
-The dataset and measurement framework (15k user evaluations) represent a valuable community resource
-In my opinion, the work is highly relevant for the growing field of interactive AI agents
minus:
-However, the practical impact is somewhat diminished by the methodological concerns around selection bias and nested data handling

Originality
plus:
-The authors run an 'In-the-wild evaluation' at a huge scale, which is rare for human-agent collaboration
-Overall, the focus on human-agent interaction (vs. agent-only) fills an important gap
minus:
-Core ML prediction approach appears standard, where the novelty primarily lies in the empirical application, domain and scale

Typos:
-Line 29 (right column):  ‘on the one hand’ without ‘on the other hand’
-Line 44 (right column): ‘a three-step framework for measuring the effect of a proposed agent chang’ is it change?
-Duplicate: Sun et al 2021

Summary of suggestions for the authors:
1. I would suggest restructuring the introduction to separate theoretical contributions from implementation details, as it was sometimes really hard to follow
2. Reorder results to prioritize PULSE framework evaluation, as you report many results that would, for example, benefit from structuring based on significance
3. I would suggest integrating techniques to account for biased and nested data structure, for example, by adding cluster-robust standard errors or mixed-effects modeling

---

> ### Author Rebuttal · Authors · 2026-03-30
>
> We thank the reviewer for highlighting that we “address a genuinely important problem of evaluating agents in realistic human-agent collaboration settings” and for recognizing that our “large-scale deployment, which is rare in academia, strengthens empirical validity.”
>
> **Weakness (soundness): We address individual questions below and will incorporate our responses into the final version.**
>
> - *Which features were extracted?* We provide the full list of features in Appendix A.2.
> - *Predictive methods "outperform baselines", but by how much?* We will clarify this in the revision to say “predictive methods improve correlation with outcomes by at least 26%  compared to baseline.”
> - *What specific problems do dialogue-specialized LLMs encounter?* Across all three LLMs evaluated, we find that LLMs tend to be more pessimistic than users. For example, o3 rarely gives a score of 5. In the final version, we will include visualizations in the Appendix that show the distribution of scores for each model.
> - *“Only 5% of sessions received feedback, likely skewing toward users with stronger opinions.”* We empirically compared labeled and unlabeled trajectories across all 15 features used in our predictive model and quantified differences using rank-biserial correlation. Overall, we found that only the user message count exhibits a moderate effect, with labeled trajectories containing more user messages (RBC=0.32). All remaining differences between labeled and unlabeled trajectories are small or negligible (RBC around 0.1 or less). The primary difference in user message count suggests that explicit feedback tends to be provided by more verbose or engaged users, but unlabeled trajectories still exhibit similar patterns of agent failures and user sentiment, indicating that silent users likely experience similar issues but may not choose to provide explicit ratings.
> - *How many users contributed multiple sessions?* We find that only 12.75% of users contributed ratings in multiple sessions (and of that percentage, the majority contributed only 2 sessions). The vast majority of users only contributed one session.
> - *“Cluster-robust methods should be employed”:* Since repeated contributions are relatively uncommon, we expect user-based dependency to have minimal impact. We therefore leave this as an avenue for future work.
> - *“The observation that GPT-5 was rated significantly lower than Claude-4-sonnet is presented as a causal explanation for user behavior differences.”:* We believe there might be a misunderstanding. We did indeed conduct a controlled experiment (column 2 of Figure 4) and find that both the naive and augmented estimators show a significant difference between GPT-5 and Claude-4-sonnet satisfaction.
>
> **Weakness (presentation): The contribution section conflates technical/theoretical contributions with practical implementation details.**
>
> We thank the reviewer for the helpful suggestions to improve the readability of our work. In the final version, we will restructure the introduction and results accordingly.
>
> **Weakness (significance): In my opinion, the work is highly relevant for the growing field of interactive AI agents. However, the practical impact is somewhat diminished by the methodological concerns around selection bias and nested data handling**
>
> We appreciate the reviewer for recognizing the importance of the work. We address the concerns around selection bias and nested data handling in bullets 4-5 in the above list.
>
> **Weakness (originality): Core ML prediction approach appears standard, where the novelty primarily lies in the empirical application, domain and scale**
>
> We refer to ICML reviewing guidelines about “novelty”, which states that “originality does not necessarily require introducing an entirely new method. Rather, a work that provides novel insights by evaluating existing methods, or demonstrates improved understanding is also equally valuable.” We believe we have provided novel insights based on the reviewer’s noted strengths: “the focus on human-agent interaction (vs. agent-only) fills an important gap.”

---

> > ### Author Rebuttal · Reviewer_r5Po · 2026-04-02
> >
> > Thank you for the rebuttal. My concerns are addressed, and I have updated my score accordingly.

---

### Official Review · Reviewer_Q1Ef · 2026-03-12

**Soundness:** 3
**Presentation:** 2
**Significance:** 2
**Originality:** 3
**Overall Recommendation:** 4
**Confidence:** 4

**Summary:**

This paper describes the need for benchmarks that assess not only the accuracy performance of fully-automated LLM agents, but also the quality of human-AI interaction when collaboratively completing tasks. The authors propose a framework to conduct this type of more human-centric evaluation, and test the framework in a real-world software engineering application. In addition to findings about how changes to the underlying models affect user satisfaction, the authors demonstrate that performance results using their framework do not always correspond with performance on more standard benchmarks.

**Compliance With Llm Reviewing Policy:**

Affirmed.

**Key Questions For Authors:**

Can the symbols in Table 1 be explicitly defined? In particular the meanings of “-“ vs “~”

The in-text explanations help clarify the results, but the depiction in Figure 4 is difficult to parse. In particular, can the condition comparison with an arrow between be made clearer?

The description of features like user sentiment and volume of user messages as primary predictors of user satisfaction seems like it’s missing a deeper discussion of what could be driving the negative sentiment/tendency to stop engaging. Otherwise it just seems like a behavioral marker of satisfaction predicting an explicit self-reported rating of satisfaction.

**Limitations:**

Using human satisfaction ratings as the primary outcome variable is a fairly limiting approach. There is utility to understanding human preferences, but preferences and performance are not always related, and humans are often limited in their metacognition. For example, human self-assessment is often unreliable, and people often struggle to explicitly describe the reasoning for their decisions. It would be useful to assess the human side of human-agent interactions with more objective, and not just subjective and self-reported metrics.

**Strengths And Weaknesses:**

The paper addresses a useful research area; namely, how best to assess not just LLM performance, but also the human-AI interaction in collaborative tasks. The human side of the equation is often simplified/abstracted or neglected entirely in this domain. However, the exclusive focus on user satisfaction to assess interactions is somewhat limiting.

The discussion of some of the factors that predict user satisfaction seems to be somewhat circular and not especially informative (see questions below).

The authors detail their methods clearly, but the depiction of results is somewhat confusing in places (see questions below).

---

> ### Author Rebuttal · Authors · 2026-03-30
>
> We thank the reviewer for highlighting that our work “addresses a useful research area” and are glad to see the recognition that the community should “assess not just LLM performance, but also the human-AI interaction in collaborative tasks.”
>
> **W1: It’s missing a deeper discussion of what could be driving the negative sentiment/tendency to stop engaging.**
>
> We manually inspected the event stream of a sample of 20 sessions that had low ratings and found that features like user sentiment and volume of user messages are often downstream of concrete interaction failures rather than purely attitudinal signals. For example, in these low-rated sessions, we observed that there are repeated errors like failed tests/CI, missing dependencies, and port/health‑check failures. In these sample sessions, we see that the user can get increasingly frustrated (based on the tone and content of their messages) after spending multiple turns trying to recover from these failures. This analysis suggests that higher message volume and negative sentiment frequently reflect interaction friction, which then drives satisfaction down. We will add examples of these failure modes to the Appendix of the final version.
>
> **W2: Using human satisfaction ratings as the primary outcome variable is a fairly limiting approach.**
>
> We note that collecting subjective ratings is a long-standing practice in the HCI literature to measure system usability [1]. We will expand the Limitations and Future Work section to include opportunities to go beyond human ratings.
>
> We conduct additional analysis to correlate user satisfaction and more objective interaction metrics (e.g., git actions). We find that satisfaction is positively correlated with git push (r=0.117, p<0.001) and git commit (r=0.101, p<0.001), and near zero for other git actions. While these effects are small, they indicate alignment between subjective ratings and objective outcomes.
>
> [1] Current practice in measuring usability: Challenges to usability studies and research. International Journal of Human-Computer Studies (2006).
>
> **W3: Presentation comments**
>
> In the final version, we will also resolve the presentation issues raised by the reviewer to define the symbols in Table 1 (where “-” means not applicable and “~” means partial support) and improve the visualization of Figure 4.

---

> > ### Author Rebuttal · Reviewer_Q1Ef · 2026-04-02
> >
> > Thank you for the responses. I believe the original score is still appropriate.

---

### Decision · Program_Chairs · 2026-04-30

**Decision:**

Accept (regular)

**Comment:**

The paper studies how to rigorously assess human-agent interactions. The paper collects user feedback, trains a model to predict user satisfaction, and uses this to build confidence intervals. They test their framework in the wild in a large-scale deployment in software engineering. They find that model choice more significantly impact user satisfaction than scaffolding choice, and they show that their framework provides more meaningful insights than typical benchmarks.

All of the reviewers appreciated the practical importance of evaluating human-AI interaction which is not captured by benchmarks. The reviewers particularly appreciated that paper performed a large-scale (15k users), in-the-wild study. Thus, I recommend acceptance. The author rebuttal addressed several of the reviewer questions, and we encourage the authors to revise the paper to integrate this feedback into the final version of the paper.